# Inflammatory Bowel Disease: A Review of Pre-Clinical Murine Models of Human Disease

**DOI:** 10.3390/ijms23169344

**Published:** 2022-08-19

**Authors:** Brunette Katsandegwaza, William Horsnell, Katherine Smith

**Affiliations:** 1Department of Infectious and Parasitic Diseases, University of Liege, 4000 Liege, Belgium; 2Division of Immunology, Department of Pathology, University of Cape Town, Cape Town 7925, South Africa; 3School of Medicine, Cardiff University, Cardiff CF14 3XN, UK

**Keywords:** inflammatory bowel disease, ulcerative colitis, Crohn’s disease, murine models, therapeutics

## Abstract

Crohn’s disease (CD) and ulcerative colitis (UC) are both highly inflammatory diseases of the gastrointestinal tract, collectively known as inflammatory bowel disease (IBD). Although the cause of IBD is still unclear, several experimental IBD murine models have enabled researchers to make great inroads into understanding human IBD pathology. Here, we discuss the current pre-clinical experimental murine models for human IBD, including the chemical-induced trinitrobenzene sulfonic acid (TNBS) model, oxazolone and dextran sulphate sodium (DSS) models, the gene-deficient I-kappa-B kinase gamma (Iκκ-γ) and interleukin(IL)-10 models, and the CD4^+^ T-cell transfer model. We offer a comprehensive review of how these models have been used to dissect the etiopathogenesis of disease, alongside their limitations. Furthermore, the way in which this knowledge has led to the translation of experimental findings into novel clinical therapeutics is also discussed.

## 1. Introduction

The inflammatory bowel diseases (IBD) ulcerative colitis (UC) and Crohn’s disease (CD) are chronic relapsing inflammatory conditions of the gastro-intestinal tract. A systematic review of bibliographic databases up to and including 31 December 2016 revealed that the reported prevalence of IBD exceeded 0.3% in North America, Oceania, and many countries in Europe [1], with analysis of data from the Global Burden of Diseases, Injuries, and Risk Factors Study (GBD) 2017 revealing an age-standardised prevalence rate of 464.5 per 100,000 population (0.46%) in the United States of America and 449.6 per 100,000 (0.45%) in the United Kingdom between 1990 and 2017 [2]. Although historically lower in East and South Asia, Oceania, and sub-Saharan Africa, the age-standardised prevalence rate of IBD has increased in newly industrialised countries over the last 30 years [2]. Currently, diagnosis of IBD is complex and invasive, effective treatment requires tailoring to the individual, and there is no cure for disease. The annual financial burden of disease estimated at more than USD 31 billion in the USA alone [3]—the increasing costs of care for disease, as well as the increasing incidence of IBD, necessitates a pressing need for cost-effective strategies to address this disease burden. The use of pre-clinical murine models of human IBD is currently employed by researchers to better understand disease etiology. Murine experimental models of human IBD exhibit immune pathological signatures resembling Crohn’s disease (CD) or ulcerative colitis (UC). These models include the chemical-induced trinitrobenzene sulfonic acid (TNBS) model, oxazolone and dextran sulfate sodium (DSS) models, the gene-deficient I-kappa-B kinase gamma (Iκκ-γ) and interleukin(IL)-10 models, and the CD4^+^ T-cell transfer model [4,5,6,7,8,9]. Although most pre-clinical murine models do not fully recapitulate the complexity of human IBD, these models have added to our knowledge about the causes of disease and have provided targets for developing new treatments. In this review, we discuss the most widely used pre-clinical murine models of human IBD, those which have proven significant in understanding disease and in the use of novel therapeutics in the clinic. Table 1 summarises the advantages and disadvantages of these pre-clinical murine models for IBD, grouped by method of induction.

## 2. Chemical Induced Colitis

### 2.1. Oxazolone Colitis

Oxazolone colitis (OC) is induced by the intrarectal administration of a haptenising agent known as oxazolone (4-ethoxymethylene-2-phenyl-2-oxazolin-5-one) in ethanol (Figure 1) [4].

Two methods of OC induction are currently employed to elicit either a self-resolving acute or a chronic response. The former is achieved by the single administration of an oxazolone enema, while the latter is preceded by a dermal pre-sensitisation five days before administration of an oxazolone enema, leading to chronic OC [10,11]. Both acute and chronic OC in mice are characterised by a superficial inflammation of the distal colon mucosa, a high volume of inflammatory infiltrate (neutrophils, macrophages, and lymphocytes), goblet-cell depletion, oedema formation, epithelial cell loss, haemorrhage, and vascular dilation [10,12,13]. These histologic features, as well as the distribution of OC, resemble human UC [4].

Chronic activation of type-2 immune responses has been proposed to be the main driver of human UC [14] (Figure 1). In the murine model, oxazolone activation of the type-2 immune response is characterised by the production of interleukin (IL)-4 in the acute phase, then superseded by the production of IL-13 in the chronic phase [10,15,16,17]. Increased production of IL-13 (and IL-5) was also evident in lamina propria T cells isolated from UC patients [18]. Interestingly, the use of mice deficient in IL-4 receptor-alpha (IL-4Rα), a common receptor for both IL-4 and IL-13, resulted in the exacerbation of OC, which could only be rescued by depletion of IL-13 in these mice [19]. Moreover, OC was prevented in mice given an IL-13 receptor subunit alpha-2 (IL-13Rα2) fusion protein that neutralises IL-13 bioactivity [10], alluding to the possible involvement of IL-13 signalling in the exacerbation of IBD in both pre-clinical models and patients [20]. Similarly, human subjects carrying a variant of IL-13 (R130Q) with a reduced affinity for IL-13Rα2 had a lower risk for CD [20]. Heller et al. proposed that prevention of OC in mice given the IL-13Rα2 fusion protein may be due to the involvement of IL-13 in mediating tissue injury and subsequent inflammation in the mucosa [10]. Subsequently, IL-13 signalling through IL-13Rα2 was shown to regulate transforming growth factor (TGF)-β1 production during OC and promoted the deposition of collagen in a lung fibrosis model [21]. However, pre-clinical experiments using Pitrakinra, an antibody that blocks the production of IL-13 and IL-4, had no significant therapeutic effect on OC [22]. No significant therapeutic effect was found in a Phase IIa clinical trial of UC patients with an antibody targeting IL-13 (anrukinzumab) [23]. Similarly, phase IIa clinical trials of UC patients with tralokinumab, another IL-13-neutralising antibody, did not significantly improve clinical response but did result in a higher clinical remission rate and mucosal healing rate in 43 patients with UC, compared to 37 patients with UC receiving a placebo control [24].

Although the initial cell source of IL-13 in UC is yet undetermined, both natural killer T cells (NKT) and conventional CD4^+^ T cells secrete copious amounts of IL-13 during OC [10,19]. CD1d-restricted NKT cells expressing an invariant T-cell receptor can secrete IFN-γ or IL-4, activating CD4^+^ T cells to become Th1 or Th2 cells (Figure 1) [25]. Indeed, oxazolone administration resulted in IL-13 production by populations of CD1-restricted invariant NKT cells [10] (Figure 1). Although depletion of NK1.1-positive cells protected mice from OC [10], removing DX5^+^CD1d-tetramer^+^ NKT cells from transferred CD4^+^ T cells still resulted in restoration of OC in mice lacking IL-4Rα on CD4^+^ T-cells, strongly suggesting that IL-4Rα-responsive Th2 cells also play a role in the development of disease [26]. Interestingly, tissue from human ulcerative colitis patients is dominated by non-classical NKT cells, differing from the dominance of invariant NKT cells in mice [18]. This discrepancy may suggest the challenges of translating animal findings to human clinical settings but may also signify the need for novel methods to define all subclasses of NKT cells in both experimental and clinical colitis.

More recently, the role of innate lymphoid cells (ILCs) has grabbed the attention of researchers seeking to elucidate the initial responders to assault on the mucosa (Figure 1). In response to IL-25 production during OC, both NKT and the nuocyte population of ILC2s produced IL-13 [27]. Interestingly, the frequency of group 2 ILC (ILC2) populations expressing chemoattractant receptor-homologous molecule expressed on Th2 cells (CRTH2) in intestinal biopsies was increased in patients newly diagnosed with UC, whereas the frequency of both ILC2 and group 1 ILC (ILC1) populations was increased in patients with established UC or CD [28]. A recent study reported increased proportions of ILC2s within the ileal fraction of CD patients, compared to healthy controls [29].

Several experimental challenges are associated with this pre-clinical model, including rapid weight loss and high levels of mortality following oxazolone administration in the acute model of disease [4]. In addition, administration of lower doses of oxazolone resulted in a dichotomous inflammatory response, characterised by the majority of mice dying rapidly, or selective survival of a minority of mice with only mild, transient disease [10]. In comparison, animals in the chronic model of OC given this same dose of oxazolone developed a sustained weight loss, with high mortality by day 9 post-administration [10]. Variation in responses and the need for optimisation in this model necessitates the use of large experimental mice numbers per treatment group, which may be costly and raise ethical issues.

As a model for human UC, oxazolone colitis aetiology does not fully reflect that in UC, exemplified by the difference in NKT subsets pivotal in both pathologies and the insignificance of IL-13 neutralisation in human UC. Although the chronic oxazolone colitis model leads to progressive colitis [10], human disease is characterised by periods of remission and repeated relapses. Researchers have broadened the use of the oxazolone colitis model in a novel humanised mouse model, transplanting human peripheral blood mononuclear cells from patients with UC into immune-deficient non-obese diabetic-severe combined immunodeficiency interleukin-2Rγ(null) mice (NOD-SCID IL2Rγ(null)), allowing for a more accurate assessment of the efficacy of therapeutics targeting the immunological responses underlying the development of IBD in patients [30].

### 2.2. TNBS-Induced Colitis

This pre-clinical model also utilises intrarectal administration of a haptenising agent in ethanol: 2,4,6-trinitrobenzene sulfonic acid (TNBS). Administration of 0.5 mg of TNBS in 50% ethanol to mice resulted in chronic transmural colitis, characterised by diarrhoea, weight loss, and rectal prolapse, pathology that mimics some characteristics of CD in humans [7]. Although one administration of TNBS resulted in acute chemical damage to the gut epithelium, inflammation was self-limiting, rather than the chronic inflammation seen in human disease [7]. Differing responses in mice are apparent in this acute model of colitis, varying according to several factors including age, genetic background, and TNBS dose [31]. To achieve chronic colitis, this model was developed by pre-sensitising the skin with 1% TNBS, followed by up to six repeated weekly intrarectal administrations of increasing doses of TNBS [17]. This model resembled the chronic phase of CD and was accompanied by production of IL-23 and IL-17 by lamina propria cells [32].

Isolated lamina propria CD4^+^ T cells from mice given TNBS secreted high levels of the Th1 cytokine interferon (IFN)-γ, resembling the cytokine profile produced by isolated lamina propria CD4^+^ T cells from CD patients [33]. This distinguished them from the Th2 profile of the same cells isolated from UC patients, or mice given oxazolone [33]. Although antibodies against IL-12, a pivotal cytokine for Th1 differentiation, abrogated established colitis and the initiation of TNBS-disease in BALB/c mice [7], TNBS-dependent colitis was exacerbated in IFN-γ-deficient mice on a BALB/c background [34].

The clinical importance of the TNBS model is demonstrated by the translation of Neurath’s anti-IL-12 antibody findings from TNBS murine experiments to successful human trials [35]. Importantly, the antibody used in these same studies was later found to react with the promiscuous p40 subunit shared by both IL-12 (a 70 kDa heterodimer of the p40 and a p35 subunit) and IL-23 (heterodimer of the p40 and a p19 subunit) (reviewed in [36]). Clinical trials with “brakinumab”, a monoclonal antibody recognising the human p40 subunit, downregulated both IL-12p70 and IL-23 secretion [37] and resulted in a clinical improvement in patients with active CD [38]. Drug development of brakinumab was discontinued due to the existence of another IL-12/IL-23 inhibitor, Stelera (ustekinumab), on the market, which significantly increased the induction and maintenance of clinical remission in patients with UC [39]. Despite the findings of exacerbated TNBS-colitis in mice lacking the p19 subunit of IL-23, novel therapies targeting IL-23 and the IL-23R have been developed and deemed successful when tested in clinical trials of patients with IBD [40].

The TNBS-colitis model has also been proven useful in understanding the mechanisms by which genetic mutations might result in IBD. This is exemplified by nucleotide-binding and oligomerisation-domain-containing 2 (NOD2)-deficient mice, designed to model one of three possible variations in the *NOD2* gene, carried by approximately 40% of CD patients [41]. Although these mice did not develop spontaneous colitis, they displayed increased susceptibility to TNBS-colitis [42]. Following administration of TNBS, *NOD2*^−/−^ mice exhibited increased lymphoid formations as well as increased IFN-γ production, paracellular permeability, and bacterial translocation in Peyer’s patches when compared to wild-type mice [43]. Studies demonstrating the absence of bacterial muramyl dipeptide recognition by macrophages from *NOD2*^−/−^ mice [44] and the increased susceptibility of these animals to bacterial-induced intestinal inflammation [45] led to the suggestion that *NOD2* plays a role in limiting inflammation in response to the intestinal bacterial flora. Moreover, a parallel study demonstrated that *NOD2* negatively regulated Toll-like receptor 2 (TLR2)-driven activation of excessive type 1 cytokine responses [46]. Indeed, the construction of mice that overexpress *NOD2* demonstrated their resistance to TLR2 ligation as well as a significantly reduced resistance to TNBS-colitis [47].

A lack of spontaneous colitis in *NOD2*^−/−^ mice was also associated with increased proportions of CD4^+^ regulatory cell populations bearing the latency associated peptide (LAP) within the lamina propria, when compared with wild-type mice [42]. Indeed, *NOD2*^−/−^ recipients of lamina propria mononuclear cells (LPMC) from ethanol-treated *NOD2*^−/−^ donors that had been depleted of CD4^+^LAP^+^Foxp3^−^ cells exhibited increased TNBS-induced weight loss when compared to *NOD2*^−/−^ recipients of intact LPMCs, which did not develop colitis [42]. Interestingly, through co-housing experiments with *NOD2*^−/−^ and wild-type mice, Amendola et al. demonstrated an influence of the endogenous gut microbiota on the proportion of LAP^+^ T cells in the lamina propria and susceptibility to TNBS colitis [42]. During co-housing with *NOD2*^−/−^ mice, acquisition of *Rickenellaceae* by wild-type mice was associated with increased proportions of CD4^+^LAP^+^Foxp3^−^ T cells and less severe TNBS-colitis [48]. The presence of *NOD2* variants in 15% of the healthy population and the failure of *NOD2*-deficient mice to develop spontaneous colitis confirmed the multigenic nature of CD [41]. However, these findings using the TNBS-colitis model strongly suggest that environmental factors impact genetic susceptibility to IBD via their impact on the microbiota. Indeed, there is now a focus on associating IBD implicated genes with variants in the gut microbiota to identify possible mechanisms by which host–gut microbial interactions influence human disease [49,50,51].

Like the oxazolone model, variation in responses to TNBS necessitates optimisation and the need to use large experimental mouse numbers per treatment group [31,52]. Variation in responses can arise due to the differing colitogenic activity of TNBS lots, TNBS dose, mouse strain, and the microbial status of the animal facility [17]. In addition, the high rates of mortality that can occur in susceptible strains [31], or following repeated administrations of TNBS [53], raise ethical implications over the use of this pre-clinical model of disease.

### 2.3. Dextran-Sulphate-Sodium-Induced Colitis

Dextran sulphate sodium (DSS) colitis is the most widely used experimental murine model of colitis, established by Okayasu in 1990 through the administration of DSS with a molecular weight of 40–50 kDa in drinking water [8]. DSS is thought to form nano-lipid vesicles with medium-chain fatty acids (MCFAs) in the colon, which fuse with colonocyte membranes and increase inflammatory cytokine levels [54]. A high-fat diet rich in MCFAs exacerbated weight loss, inflammatory cytokine expression, and colon shortening in this model, with dodecanoic acid favouring disruption of intestinal barrier function and increased vesicle formation in vitro [54]. One day after administration, DSS particles were present systemically in Kupffer cells of the liver, in macrophages of the mesenteric lymph node, and in the lamina propria of the large intestine [55]. DSS administration was also characterised by erosion of the intestinal epithelium, inflammatory infiltration of the large intestine, and dysbiosis of the intestinal microbiome [8]. While these features are similar to those found in human disease, the transmural inflammation apparent in TNBS-colitis is absent in this model [56]. Although repeated rounds of DSS can be administered to provide the pattern of remitting, relapsing inflammation in human IBD, some of the limitations of this model include inter-batch variability of DSS and the need to optimise DSS dose, given the impact of the intestinal microbiome on disease [56,57].

The acute inflammatory response in DSS-colitis was typified by increased expression of IL-1β, IL-6, and tumour necrosis factor (TNF)-α in the colon (Figure 2) [58].

Although prototypical Th2-type biased BALB/c mice did not develop chronic colitis, disease was characterised by Th1 and Th2 cytokines [59] and was absent in prototypical Th1-type C57BL/6 mice lacking the Th2 cytokine IL-4 [60]. Using this model, plasma levels of haptoglobin were found to correlate well with human clinical symptoms of IBD [61].

DSS-colitis can be induced in immunodeficient mice including recombination-activating gene (RAG)-1-deficient and severe combined immune deficient (SCID) mice, suggesting the dispensability of the adaptive immune system in initiating disease [62,63]. Although Kim et al. demonstrated colitis induction in RAG-1-deficient mice, the resultant mild colitis in these mice compared to their wild-type counterparts insinuates that lymphocytes may be necessary for subsequent colitis progression [63]. Histological assessment of biopsy specimens from IBD patients correlated UC and CD with severe mononuclear cell infiltration and basal plasmacytosis (plasma B cells) [64]. In the acute phase of the DSS-colitis model, this infiltrate consisted of innate macrophage, neutrophil, and eosinophil populations recruited following increased cytokine and chemokine expression [60].

Elevated expression of IL-17 and IL-23 was reported in IBD patients and in DSS-colitis, where expression of the two cytokines was intertwined (Figure 2) [37,65,66,67,68,69]. The use of the DSS-colitis model to test the role of these cytokines in disease has revolutionised not only the possible interventions available for patients but has also developed our understanding of mucosal immunology. At steady state, the p19 subunit of IL-23 is highly expressed within Peyer’s patches and the thymus, as well as in polarised Th1 cells, activated macrophages, and dendritic cell populations derived from peripheral blood [70]. IL-23 induced proliferation of memory T cells and elevated secretion of IL-17 in vitro [71]. Subsequently, IL-23 signalling within intestinal epithelial cells was found to play an important role in protection against DSS colitis by regulating regenerating-islet-derived protein 3-beta (Reg3β)-dependent control of flagellated intestinal bacterial abundance and promoting IL-22 production [72]. Indeed DSS-colitis was exacerbated in IL-22-deficient mice [73] and blockade of IL-22 expression delayed recovery from DSS-colitis and exacerbated disease scores [74]. 

Interestingly, the IL-23 receptor was upregulated in lamina propria CD4^+^ T cells from both UC and CD patients [75]. While upregulated IL-23p19 mRNA expression correlated with increased IL-17 mRNA expression in UC patients, it correlated with increased IFN-γ mRNA expression in CD patients [75]. Both the IL-17 family members IL-17A and IL-17F co-ordinate local tissue inflammation by regulating the release of proinflammatory cytokines and the recruitment of neutrophils [76]. Thus, activation of the transcription factor retinoid-acid-related orphan nuclear receptor gamma (RORγt) in CD4^+^ T helper 17 (Th17) cells led to the production of IL-17, IL-12, IL-21, and IL-23 as well as the upregulation of IL-23R (reviewed in [77]) (Figure 2). In vitro culture of CD and UC mucosal tissue demonstrated enhanced production of IL-21, with IL-21 inhibition attenuating intestinal inflammation as well as the Th17 response in DSS-exposed mice [78].

However, DSS-colitis was shown to be independent of the production of IL-17 within lamina propria CD4^+^ T cells [79], where neutralisation of IL-17A in prototypical Th2 biased BALB/c mice aggravated disease [80]. Subsequent studies demonstrated that production of IL-17A by specialised γδ^+^CD4^+^ T cells was shown to regulate intestinal permeability and injury in the DSS model, independently of IL-23 signalling [81]. Paradoxically, although one study had demonstrated that IL-17A-deficiency in prototypical Th1-type C57BL/6 mice reduced DSS colitis [82], a separate study using IL-17A- and IL-17F-deficient mice demonstrated a deleterious role for IL-17F in the development of disease [83]. In this latter study, both T-cell-derived IL-17F and colonic-resident-cell-derived IL-17F controlled antimicrobial-protein-dependant changes to the commensal microbiota, as well as the expansion of anti-inflammatory regulatory T cell populations in the colon [83].

In UC patients and the DSS model, high levels of IL-17F were associated with an abundance of *Fusobacterium nucleatum*, an opportunistic bacterium that enhanced pro-inflammatory cytokine production and disrupted the mucosal barrier via the nuclear factor kappa-light-chain-enhancer of activated B cells (NF-kB) pathway [84]. These studies explained the poor efficacy of anti-IL-17A monoclonal antibody and anti-IL-17 receptor monoclonal antibody treatment of CD patients, both of which worsened disease [85,86]. Although neutralisation of IL-17A and IL-17F with bimekizumab appeared efficacious in alleviating tissue inflammation in patients with mild or moderate-to-severe psoriasis [87,88], it is yet to be tested in clinical trials with IBD patients.

A potent pro-inflammatory cytokine, TNF-α, was highly upregulated during DSS-colitis and was elevated in intestinal biopsies from CD patients [89]. Mice engineered to express elevated TNF-α levels (TNF^DARE^ mice) developed spontaneous intestinal inflammation resembling human CD [90]. Although neutralisation of TNF-α had detrimental effects on inflammation during acute DSS-induced colitis, it significantly improved histological scores in chronic DSS-induced colitis [91]. TNF-α was later proposed to regulate epithelial cell apoptosis, neutrophil infiltration, pro-inflammatory cytokine production, and systemic inflammation in response to the intestinal microbiota through TNF-α receptor 1 [92]. Since the approval of the anti-TNFα drug infliximab for treatment of CD in 1998 and subsequent approval for use in UC in 2006, anti-TNF-α therapy has become the cornerstone of treatment strategies for IBD. One single administration of infliximab induced clinical remission in 48% of CD patients, compared to 4% in the placebo group [93]. However, treatment failures with anti-TNF-α are common [94], and there have been concerns with potential complications from therapy, including an increased risk of malignancies [95]. Interestingly, a recent study correlated the responsiveness of IBD patients to TNF-α therapy with bacteria diversity, but not fungal diversity, in the faecal microbiome [96].

## 3. Spontaneous Colitis

Several murine models of spontaneous colitis have also been used to model IBD triggered by dysbiosis in the gut microbiota and specific IBD-related genes, reviewed in brief below.

### 3.1. Iκκ-γ (NEMO) Deficiency Colitis

Conditional ablation of the NF-kB essential modulator (NEMO), also known as I-kappa-B kinase gamma (Iκκ-γ), within the intestinal epithelium, resulted in spontaneous colitis in mice [6]. Chronic disease in intestinal-epithelial-cell-specific NEMO-deficient mice was associated with TNFR1-dependent colonic epithelial cell death, compromised epithelial integrity, bacterial translocation in the colon, immune cell infiltration, and increased expression of pro-inflammatory cytokines including TNF-α [6]. Absence of disease in double-deficient (NEMO^IEC−KO^ + MYD88^−/−^) mice, lacking NEMO and the important bacterial sensor myeloid differentiation primary response 88 (MYD88), supported a role for the gut microbiota in driving colitis [6]. Indeed NEMO^IEC−KO^ mice raised under germ-free conditions did not develop spontaneous colitis, whereas co-housing of these mice with specific pathogen-free animals restored disease [97]. Two-week-old NEMO^IEC−KO^ depleted of beta-defensin-3, an antimicrobial peptide (AMP) produced by intestinal Paneth cells following bacterial sensing [6], developed spontaneous colitis. These findings supported genome-wide DNA copy number profiling studies, which associated a lower copy number of the human homologue of this murine defensin, human beta defensin-2, with a higher risk of colonic CD [98].

Ablation of the NF-κB p65 subunit, encoded by the RELA gene, resulted in a loss of Paneth cells but an absence of spontaneous colitis, suggesting that the epithelial protective effects of NEMO signalling may be NF-κβ independent [99]. These authors suggested a therapeutic potential for receptor-interacting serine/threonine-protein kinase 1 (RIPK1) inhibition in IBD patients, after demonstrating that RIPK1 prevented epithelial cell death and colitis development in NEMO^IEC−KO^ mice [99]. Although this pre-clinical model is of use to study the interactions between the intestinal microbiome and spontaneous colitis, NEMO-deficient patients exhibit a variety of mucocutaneous diseases, and it is feasible that they may also carry concomitant mutations in IBD-associated genes [100].

### 3.2. Interleukin-10 (IL-10) Deficiency Colitis

Although IBD is common among adults, childhood IBD constitutes about a quarter of all patients with IBD [101]. Approximately 15% of childhood IBD occurs in children <6 years old and is termed very early onset IBD (VEO-IBD) [102]. While most childhood IBD cases are polygenic in nature, many children with VEO-IBD have an underlying monogenetic disorder that results in severe enterocolitis, including mutations in IL-10 and/or IL-10R (reviewed in [103]). Mice deficient in IL-10 develop spontaneous enterocolitis, characterised by progressive cellular infiltration of the cecum, colon, rectum, and small intestine, with transmural lesions and a high incidence of colorectal adenocarcinomas observed in 6-month-old mice [5,104]. Mice lacking IL-10 receptor β also developed spontaneous enterocolitis [105,106]. However, this receptor is shared with other type II cytokine receptors including those specific for IL-22, IL-26, and lambda interferons, in addition to IL-10 [107].

Interestingly, IL-23 deficiency (IL-12p19^−/−^), but not IL-12 deficiency (IL-12p35^−/−^), abrogated spontaneous colitis in IL-10^−/−^ mice [108]. These findings demonstrate the importance of IL-23 instead of IL-12 in the development of experimental IL-10^−/−^ colitis. As previously discussed (under DSS colitis), IL-23 synergises with IL-17, IL-21, and IL-22 to drive colon inflammation in the event of defective immune regulation mechanisms. Work from this group suggested a role for IL-23-dependent production of IL-17 and IL-6 in accelerating IBD in a T cell transfer model (reviewed below) [108]. Subsequently, IL-22 was found to be frequently co-expressed with IL-17A, with the cellular source of IL-22 playing an important role in determining a role in host defence, or disease pathogenesis (reviewed in [109]). IL-10 inhibited IL-22 secretion by macrophages and T cells [110], and spontaneous colitis was prevented in mice lacking IL-10 and IL-22 [111]. In the latter study, IL-22 was expressed in the lamina propria by CD4^+^ T cells and populations of innate lymphoid cells [111].

An important role for the gut microbiota in influencing the general enterocolitis seen in IL-10-deficient mice was implicated by a lack of disease in mice housed in specific pathogen-free conditions, or in germ-free conditions [5,112]. Treatment of IL-10-deficient mice with two different antibiotics, both shown to improve scores in patients with Crohn’s disease [113,114], attenuated the development of spontaneous colitis [115]. Similar to IBD patients, colitic IL-10-deficient mice exhibited a markedly reduced species diversity in their faecal microbiome when compared to disease-free controls [111,112,116]. In addition, IL-10/IL-22 double-deficient mice lacking colitis exhibited higher microbial diversity when compared to IL-10-deficient mice [111].

Although infection with *Helicobacter* species was found to drive enterocolitis in IL-10-deficient mice [117,118], adult germfree IL-10-deficient mice in a specific pathogen-free (SPF) environment lacking *Helicobacter* developed more severe colitis when compared to IL-10-deficient mice colonised at weaning [112]. Similarly, taxon-based analysis of the microbiota revealed that levels of the *Helicobacter* genus inversely correlated with severe colitis in IL-10-deficient mice, demonstrating disease is not dependent on co-colonisation with this pathobiont [111]. Mono-colonisation of germfree IL-10-deficient mice with *Enterococcus faecalis* triggered enterocolitis [119], with subsequent studies showing distal colitis in *Enterococcus faecalis* mono-colonised IL-10-deficient mice and a mild cecal inflammation in *Escherichia coli* mono-colonised animals [120]. Administration of *Lactobacillus* species from birth prevented the development of enterocolitis in IL-10-deficient mice [121] and administration of an 8-strain probiotic to IL-10-deficient mice with established inflammation improved disease by normalising the epithelial barrier [122]. Although antibiotic treatment provides an attractive candidate for the treatment of colitis in patients with IL-10 deficiency, a recent study in IL-10-deficient animals demonstrated that intermittent antibiotic exposure accelerated the development of severe colitis in male mice [123].

Repeated intravenous or subcutaneous administration of recombinant human IL-10 (rhIL-10) to patients with active steroid-resistant Crohn’s disease [124] or mild to moderately active Crohn’s disease [125] was well tolerated and resulted in an improvement in disease scores, while subcutaneous administration of rhIL-10 to healthy patients and those with CD resulted in a reversible decline in haemoglobin and thrombocyte counts, or an increased production of IFN-γ [126,127,128,129]. Due to these undesirable side effects, researchers engineered *L. lactis* bacteria [130,131] and targeted the transcytosis pathway activated by the intestinal pathogen *Vibrio cholerae* with a novel fusion protein [132] to deliver IL-10 to the gastrointestinal tract. Daily intragastric delivery of L. lactis modified to contain the murine IL-10 prevented the onset of colitis in IL-10-deficient mice more efficiently than systemic delivery of recombinant murine IL-10 [130]. An absence of systemic side effects in a small phase 1 clinical trial of CD patients with L. lactis, modified to contain the human IL-10 sequence, demonstrated the safety of this approach [133]. Although the IL-10 fusion protein and an invasive *L. lactis* strain carrying the IL-10 plasmid both alleviated colitis in a range of colitis models, including IL-10-deficient mice [131,132], IBD patients can have raised levels of IL-10 in serum [134] and may not benefit from IL-10 therapy [129,135].

## 4. Immune Cell Induced Colitis

### T-Cell Adoptive Transfer Model

The transfer of murine CD45RB^high^CD4^+^ T cells from healthy donors to severe combined immunodeficiency (SCID) (Figure 3) mice resulted in the development of a lethal wasting disease, an influx of inflammatory cells, and increased inflammatory cytokine production in the colon of the recipient [9].

The resulting chronic colitis that developed within 5–8 weeks was T cell dose-dependent, and despite the presence of transferred T cells in several organs, significant pathology was limited to the large intestine [9]. Interestingly, the transfer of naïve CD45RB^high^CD4^+^ cells into SCID mice with a reduced microbiota significantly reduced colitis when compared to SCID mice housed in SPF conditions [136]. Restoration of T-cell proliferation in germ-free *RAG*-deficient mice following reconstitution with cecal bacterial lysate-pulsed dendritic cells and induction of colitis in SPF *RAG*-deficient mice following the transfer of T cells specific for a microbiota flagellin antigen cemented the role of the commensal microbiota in driving disease in this model [137].

Although both CD45RB^low^ and CD45RB^high^CD4^+^ T cells homed to the intraepithelial and lamina propria compartments of both the small and large intestine [136], CD45RB^low^ or total unfractionated CD4^+^ T-cells were incapable of inducing colitis [9]. Following the discovery that the CD45RB^low^ population comprised a population of CD4^+^CD25^+^ regulatory T cells that could cure colitis in SCID or *RAG1*-deficient recipients of CD4^+^CD45RB^high^ T cells [138], the T-cell transfer model of colitis was further refined by reconstitution of SCID mice with CD25-depleted CD4^+^ cells [139]. Although colitis in this model was accredited to the production of the Th1 cytokine IFN-γ [9,140], enhanced emergence of IL-17A^+^IFN-γ population of T cells and suppression of populations of Foxp3^+^ and IL-10-producing regulatory T cells was attributed to IL-23R signalling in T cells [141]. Similarly, an absence of IL-23 in IL-12p19-deficient *RAG*-deficient recipients resulted in significantly decreased levels of proinflammatory cytokine production in the intestine [142]. While intestinal inflammation was significantly reduced in the *RAG*-deficient recipients of IL-23R-deficient CD45RB^high^CD4^+^ T cells, or p19^−/−^RAG^−/−^ recipients of CD4^+^CD25^−^CD45RB^high^ T cells, these mice still exhibited systemic inflammation and weight loss [141,142], suggesting a tissue-restricted activity of IL-23.

Using IFN-γ-deficient RAG^−/−^ recipients, IFN-γ was subsequently shown to be dispensable for the induction and progression of CD4^+^CD25^−^CD45RB^high^ T cell transfer colitis [143]. Inflammatory responses in these recipients favoured IL-17A production, where neutralisation of IL-17A/F significantly reduced weight loss and histopathological score in these animals [143]. A population of CD4^+^IL-17F^+^ T cells was sufficient to induce colitis in *RAG1*-deficient recipients, where the transition of these precursors to Th1-like cells was an absolute requirement for disease [144]. Interestingly, IL-21 signalling was recently proposed to dampen T-cell transfer colitis by reducing IL-17A production and augmenting IL-22 production in populations of ILC3s [145].

Intriguingly, *RAG*-deficient (*RAG*^−/−^) recipients lacking signalling through the type 1 interferon α/β receptor subunit 1 (*RAG*^−/−^*IFNAR1*^−/−^) developed severe accelerated colitis following T-cell transfer of CD45RB^high^CD4^+^ T cells from wild-type mice [146]. Co-transfer of CD4^+^CD25^+^Foxp3^GFP+^ regulatory T cells (Treg) and CD45RB^high^CD4^+^ T cells to *RAG*^−/−^*IFNAR1*^−/−^ recipients resulted in a loss of Treg expansion, Treg stability, and suppressive function when compared to *RAG*^−/−^ recipients [146]. Kole et al. revealed an important role of IFNAR signalling on *RAG*^−/−^ hematopoietic cells to suppress colitis in this model [146]. Subsequently, NK cells were revealed as an important regulator of colitis in this system through the action of perforin on Th1 immune responses [147]. Despite this immune-regulatory role, intraepithelial NK cells were the cellular source of IFN-γ in germ-free SCID colonised with altered Schaedler’s flora and were proposed to contribute to the pathogenesis of colitis in these mice [148]. In addition, NKGD receptor signalling on NK cells, activated macrophages, activated CD8+ αβ T cells, γδ+ T cells, and subsets of NKT cells was proposed to play a role in initiating colitis [149]. Recent evidence suggests that 6-mercaptopurine, a mainstay for IBD patients to maintain disease remission, mediated apoptosis of NK cells and decreased NK cell activity and number in the intestinal biopsies of CD patients [150].

An advantage to this model is the ability to examine early immunological events associated with gut inflammation, including regulatory T cells responses. In addition, the apparent inflammatory response in both the small bowel as well as the colon makes this model ideal for the study of CD [140]. However, the SCID and *RAG*^−/−^ recipients used in this model spontaneously develop T and B cell populations as they age [151], while NK cells from *RAG*^−/−^ mice are reported to exhibit altered function and fitness [152]. The lower prevalence of human SCID patients, estimated at 1 in 58,000 new-borns in the United States (0.0017%) [153], compared to a higher incidence of CD (estimated at 10.7 cases per 100,000 individuals in the United States (0.0107%) [101], points to a factorial model of CD induction/development, independent of primary immunodeficiencies in T and B cells. This excludes this model as an exact model of CD, despite the many similarities.

## 5. Conclusions

Although several murine findings have contributed immensely to IBD research, the translation of mouse work to human applications should be cautionary to all researchers due to some reasons mentioned below:

Mice and humans differ in their immune responses due to different immune system development, immune activation, and immune responses to similar antigens [154].Mouse experiments tend to not recapitulate the genetic and environmental diversity inherent in human populations, although these can be closely modelled through diet, microbiome, and environmental manipulations.To determine the importance of certain genes in disease pathology, researchers tend to depend on transgenic knockout mice or antibody depletion, whereas human disease risk is rarely associated with the complete loss of function of a single gene or protein.Most pre-clinical murine experiments fail to account for variability in response to a therapeutic intervention that may appear in human trials due to genetic polymorphisms. This may be addressed through the use of outbred mice but may have ethical and economic implications, due to the likely need to use a large sample size to achieve statistical significance between treatment groups [155].

Despite these limitations, murine pre-clinical models still present the opportunity to discover potential mechanisms that drive IBD, if experimentation considers the different caveats associated with each model.

## Figures and Tables

**Figure 1 ijms-23-09344-f001:**
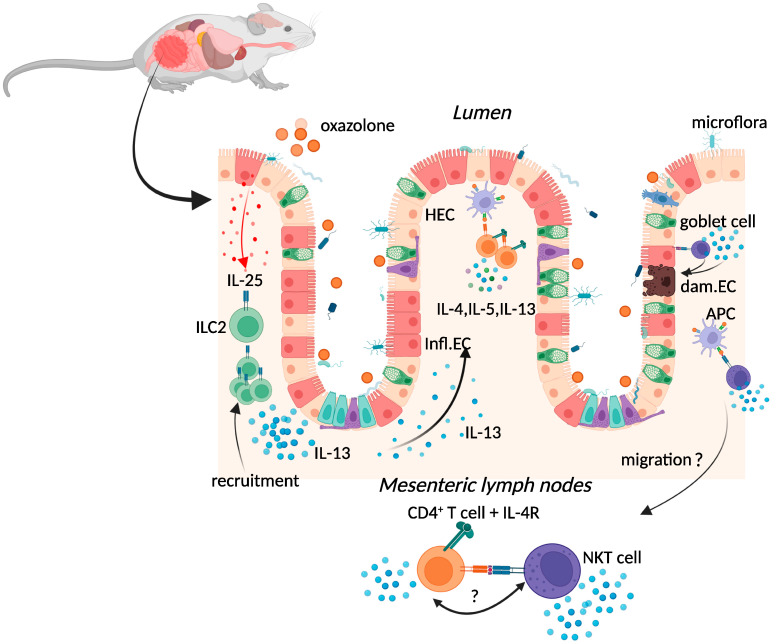
Immune response during oxazolone colitis. Oxazolone administration results in the production of IL-25, activation of ILC2, and production of IL-13, activating CD4+ T cell responses and amplifying type-2 cytokine production. Oxazolone administration also results in the expansion of T cells with surrogate markers of NKT cell function and IL-13 production by populations of CD1-restricted NKT cells. The resulting chronic inflammatory responses result in goblet cell depletion, increased intestinal permeability, and increased adhesion of commensal intestinal microbiota to the epithelium. Key: inflamed epithelial cells (Infl.EC), healthy epithelial cells (HEC), damaged epithelial cell (dam.EC). Created with BioRender.com, accessed on 18 August 2022.

**Figure 2 ijms-23-09344-f002:**
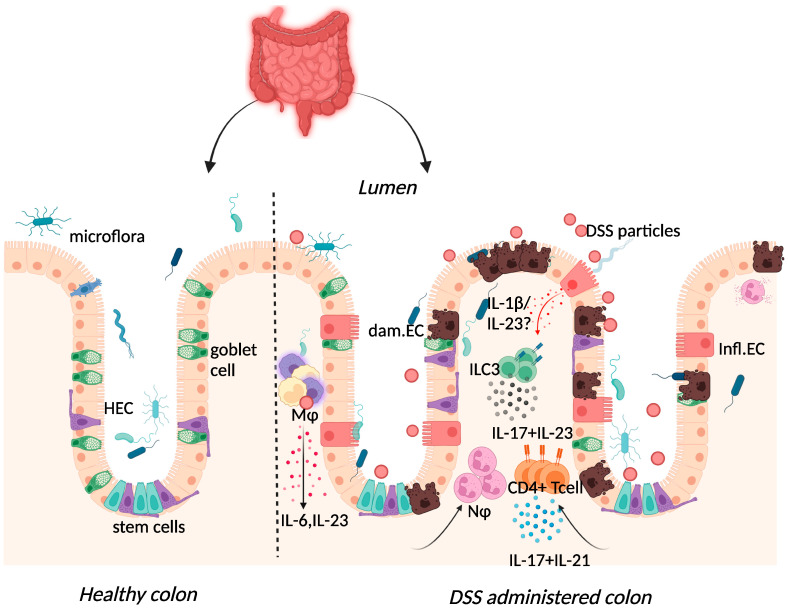
Immune response to DSS administration. DSS administration results in epithelial release of IL-1β, activation of ILC3, and release of IL-23. IL-23 release results in the influx of neutrophils and CD4+ T cells, which further respond through enhanced IL-17 signalling. The resulting chronic inflammatory responses result in goblet cell depletion, increased intestinal permeability, and increased adhesion of commensal intestinal microbiota to the epithelium. Key: Inflamed epithelial cells (Infl.EC), healthy epithelial cells (HEC), damaged epithelial cell (dam.EC), macrophages (MΦ), neutrophils (NΦ). Created with BioRender.com, accessed on 18 August 2022.

**Figure 3 ijms-23-09344-f003:**
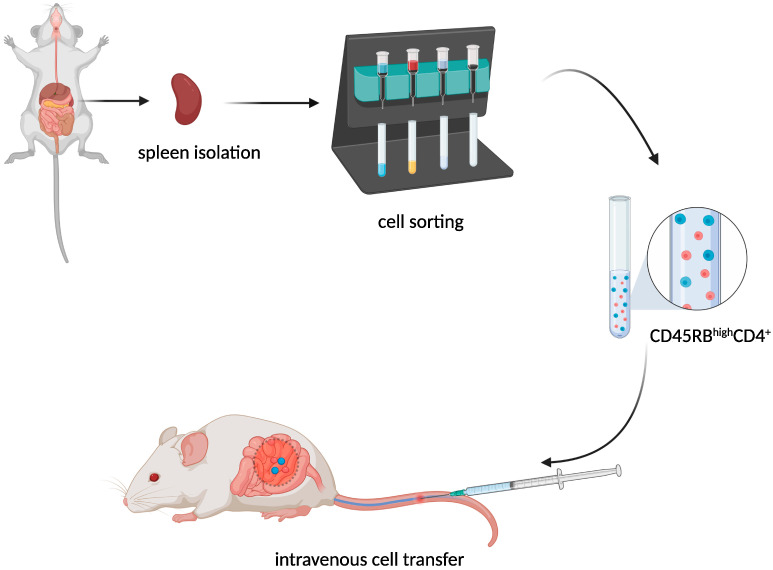
CD45RB^hi^CD4^+^ T cell adoptive transfer scheme. Created with BioRender.com, accessed on 18 August 2022.

**Table 1 ijms-23-09344-t001:** Summary of the advantages and disadvantages of the most widely used pre-clinical murine models of IBD, organised by induction method.

Induction Method	Examples	Disease Modelled	Advantages	Disadvantages
**Chemical Induced Colitis**	Oxazolone colitis	UC	Model used to uncover the role of type-2 immune responses and propose a role for NKT cells during UC. Inflammation in the distal colon closely resembles UC.	High mortality of experimental animals, or a dichotomous inflammatory response required optimisation and a large sample size. Despite findings from this model, clinical trial of novel therapeutics targeting IL-13 were not significant.
	Trinitrobenzene sulfonic acid (TNBS) colitis	CD	Model revealed importance of IL-12 and IL-23 and successful clinical trials. Model used to develop understanding of genetic determinants of CD (e.g., *NOD2*). Model enables better understanding of how host-gut microbial interactions influence disease. Mice develop a chronic transmural colitis, mimicking characteristics of CD.	Variation in responses requires optimisation and the use of large experimental mouse numbers per treatment group. Variability between TNBS lots requires testing and re-optimization between batches
	Dextran Sulfate Sodium (DSS)	UC and CD	The Th1/Th2 cytokine mileu closely models both UC and CD Model revealed the importance of innate immunity in disease induction. Model has revealed the importance of IL-17 and IL-23 in IBD.	Interbatch variability of DSS affects colitis phenotype. Disease is highly dependent on microbial status Repeated administration of DSS required to elicit chronic, remitting and relapsing colitis.
**Spontaneous colitis**	Iκκ-γ (NEMO) Deficiency Colitis	NEMO-deficiency	This model enables research into the interaction between gut microbiota and spontaneous intestinal inflammation.	NEMO-deficient patients exhibit a variety of mucocutaneous diseases and may carry concomitant mutations in IBD-associated genes
	IL-10 Deficiency Colitis	Childhood IBD	Model has enabled research into the influence of the microbiome on intestinal inflammation.	IBD patients are not always deficient in IL-10 and may not benefit from IL-10 therapy.
**Immune cell Induced Colitis**	CD4^+^ T cell transfer	CD	Revealed the importance of regulatory T cells in IBD development. Allows early events underlying IBD to be studied. Supported a role for IL-17 and IL-23 in IBD development. Supported a role for NK cells in IBD development. Inflammation in small bowel and colon similar to CD.	“Leakiness” and altered NK function in immune-deficient mice. Model requires use of magnetic cell separation and/or automated cell sorting, which can be costly. Use of immune-deficient mice precludes modelling of the multifactorial nature of CD induction/development.

## Data Availability

Not applicable.

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
