# Peer review of "Inflammatory Bowel Disease: A Review of Pre-Clinical Murine Models of Human Disease"

_ijms, 2022, doi:10.3390/ijms23169344_

Round 1
Reviewer 1 Report
The review-type manuscript by Brunette Katsandegwaza et al. is written nicely, however I have some suggestions for the Authors.
In my opinion, preclinical IBD models should first be divided into specific groups, i.e. chemically induced, genetic, immunological, and spontaneous models, and only then should the selected models be described in detail within these 4 subgroups. Then the work will be more clear and readable.
There are also no tables, e.g., a table with the division into 4 groups of preclinical models and the assignment of models to a given subgroup + literature review, what we can understand from a given model, what we can assay in a given model, disadvantages, advantages, and – what important - what new we know about a given model.
The figures (esp. 1 and 2) are good, and informative, but they could have a slightly better resolution.
Author Response
We thank the reviewer for their positive comments regarding our review. Below we respond point by point to their comments.
In my opinion, preclinical IBD models should first be divided into specific groups, i.e. chemically induced, genetic, immunological, and spontaneous models, and only then should the selected models be described in detail within these 4 subgroups. Then the work will be more clear and readable.
We agree that dividing the review into these sections will help with making it clearer and more readable. We now subdivide the review into 3 groups: "Chemical Induced Colitis", "Spontaneous Colitis" and "Immune cell Induced colitis" and renumber the review accordingly.
There are also no tables, e.g., a table with the division into 4 groups of preclinical models and the assignment of models to a given subgroup + literature review, what we can understand from a given model, what we can assay in a given model, disadvantages, advantages, and – what important - what new we know about a given model.
We also now include a table to summarise all pre-clinical models of IBD, grouped into Chemical Induced Colitis, Spontaneous Colitis and Immune cell Induced colitis, as per the restructured review (Table 1). In this table we highlight the human disease each model represents most closely, as well as the disadvantaged and advatages of each model. We have also edited the text to highlight this change as follows (Line 46-49):
"In this review we discuss the most widely used pre-clinical murine models of human IBD; those which have proven significant in understanding disease and in the use of novel therapeutics in the clinic. Table 1 summarizes the advantages and disadvantages of these pre-clinical murine models for IBD, grouped by method of induction."
The figures (esp. 1 and 2) are good, and informative, but they could have a slightly better resolution.
Thank you for this information, we have now uploaded high resolution pdf copies of each of the figures.
Reviewer 2 Report
The manuscript reports various preclinical models of intestinal inflammation. I suggest enriching the TNBS-induced colitis model with more bibliographical references. currently (lines 133-140) the review only refers to an old paper.Has the model been optimized over time?
line 134, what does “low dose” refer to? Better graphics are needed for the figures(Fig. 1 AND Fig.2).
I suggest giving more emphasis to the advantages and limitations of the various models
Author Response
We thank the reviewer for the positive review of our manusript. We address the comments point by point, as detailed herein:
The manuscript reports various preclinical models of intestinal inflammation. I suggest enriching the TNBS-induced colitis model with more bibliographical references. currently (lines 133-140) the review only refers to an old paper.
Has the model been optimized over time?
We have now enriched the bibliographical references for this section and now detail how this model has been recently developed, as well more information on the limitations of this model (Line 141-148 and Line 199-204).
line 134, what does “low dose” refer to?
We now detail the amount or percentage of TNBS used in the original and modified models, instead of the more vague descriptor of "low dose". (Line 139 and Line 146).
Better graphics are needed for the figures(Fig. 1 AND Fig.2).
Thank you for this information. We have now uploaded high resolution pdf copies of each of the figures.
I suggest giving more emphasis to the advantages and limitations of the various models
We have now included a Table, which highlights the advantages and limitations of each of the various models, grouped by "Induction Method"(Table 1).